# Myoclonus Secondary to Amantadine: Case Report and Literature Review

**Jamir Pitton Rissardo *** and **Ana Letícia Fornari Caprara**

Medicine Department, Federal University of Santa Maria, Santa Maria 97105-900, Brazil;
ana.leticia.fornari@gmail.com
* Correspondence: jamirrissardo@gmail.com

**Abstract:** The usual adverse events of amantadine are dizziness, dry mouth, and peripheral edema. Postmarketing experience has revealed abnormal movements such as tremors, involuntary muscle contractions, and gait abnormalities. Herein, we report a case of an elderly male who presented with generalized twitching associated with amantadine. A 64-year-old male presenting with jerking movements within one day of onset was admitted. Sudden and involuntary distal lower and upper limb muscle twitching was observed. The subject presented subsequent brief movements when attempting to stand or hold arms antigravity. He was diagnosed with Parkinson's disease three years ago. Eight days before the presentation to the emergency department, he consulted with his primary care physician, who prescribed amantadine to improve his motor symptoms. On the seventh day, he developed brisk abnormal movements. Laboratory exams, neuroimaging, and electroencephalogram were unremarkable. Amantadine was discontinued. After three days, the patient reported that his jerking movements had fully recovered. To the authors' knowledge, 22 individuals with amantadine-associated myoclonus had already been reported in the literature. The pathophysiology of amantadine-induced myoclonus is probably related to serotoninergic pathways. Myoclonus secondary to amantadine was slightly more common in men. The population affected was elderly, with a mean and median age of 67.7 and 64 years.

**Keywords:** Parkinson's disease; myoclonus; movement disorder; amantadine; 1-adamantylamine

## 1. Introduction

Amantadine hydrochloride was developed as an antiviral medication. One of the pioneer amantadine clinical trials showed improvement in the influenza virus infection course and motor symptoms of patients affected by Parkinson's disease. In 1973, the Food and Drug Administration approved amantadine for treating Parkinson's disease [1].

The most common adverse events of amantadine are dizziness, dry mouth, peripheral edema, and livedo reticularis. Postmarketing experience has revealed that some patients could develop abnormal movements such as tremors, involuntary muscle contractions, and gait abnormalities [2,3]. To the authors' knowledge, 22 individuals of amantadine-associated myoclonus have already been reported in the literature [2,4–18].

Herein, we present a case of an elderly male who presented with generalized twitching movements, probably secondary to amantadine. Moreover, we provide a table with all the reported cases and present a figure describing the therapeutic range based on amantadine reports and clinical trials.

## 2. Case Report

A 64-year-old male presenting with jerking movements within one day of onset was admitted to our emergency department. He reported that the abnormal involuntary movements became worse throughout the last day. His vital signs, such as temperature, heart rate, respiratory rate, and blood pressure, were within normal limits.

On neurological examination, sudden and involuntary distal lower and upper limb muscle twitching was observed. Additionally, the subject presented subsequent brief movements when attempting to stand or hold arms against gravity. Bradykinesia, tremor, and postural instability associated with shuffling gait were noted. The muscle mass and strength were normal (Grade 5—Medical Research Council). The assessment of cranial nerves was unremarkable. Deep tendon reflexes were normal and active. His neurological family history was unremarkable. Additionally, he had no history of known drug allergies or adverse drug reactions.

He was diagnosed with Parkinson's (Hoehn and Yahr stage II) three years ago. In the last month, the individual reported having more common off-periods with difficulty walking. He only used 100 mg levodopa + 25 mg benserazide tablet thrice daily. Eight days before the presentation to the emergency department, he consulted with his primary care physician, who added amantadine to improve his motor symptoms. Amantadine hydrochloride 100 mg tablet once a day for three days was started. After three days, he increased the amantadine dosage to one tablet twice daily. On the seventh day, he began with brisk abnormal movements.

Laboratory exams were within normal limits, including serum creatinine levels. A cranial computed tomography scan was normal. A brain magnetic resonance (1.5 Tesla) was normal without changes after contrast material. Cerebrospinal fluid analysis showed 60 mg/dL of glucose (98 mg/dL plasma glucose), 30 mg/dL protein, 0 leukocytes, and 0 red blood cells. An electroencephalogram was normal without background seizure activity, slowing, or suppressions.

On the second admission day, it was observed that the symptoms worsened approximately two hours after the amantadine administration (Video S1). It was hypothesized that his myoclonus was probably associated with amantadine. Clonazepam and hydration were started. Amantadine was discontinued. After three days, the patient reported that his jerking movements had fully recovered. In the long-term follow-up of one year, the patient did not have a recurrence of the myoclonus. He continued with his baseline Parkinson's symptoms and was administered 100 mg levodopa + 25 mg benserazide, one tablet, five times daily.

### 3. Discussion

The exact mechanism of amantadine for managing parkinsonism and drug-induced extrapyramidal reactions is unknown. However, five main indirect pathways have been studied to explain amantadine's involvement in the cortico-striato-pallido-thalamo-cortical loop (Figure 1).

Amantadine is believed to activate presynaptic dopamine receptor D2 reducing the dopamine transporter related to dopamine reuptake [19]. In animal studies, amantadine inhibited monoamine oxidase B, decreasing dopamine degradation [20]. Interestingly, amantadine is an agonist of the σ1 receptors, which is probably related to the psychostimulant-like effects of this compound [21]. Amantadine blocks alpha-7 nicotinic receptors, which explains its anticholinergic side effects like xerostomia, urinary retention, and constipation [22].

Amantadine may influence serotoninergic pathways, increasing serotonin availability. In rat models, this drug increases serotonin release and inhibits serotonin reuptake in presynaptic neurons [23,24]. One of the hypotheses for the pathophysiology of myoclonus is related to serotonin augmentation in the cerebellar output [25]. Therefore, myoclonus secondary to amantadine could be related to a serotonergic mechanism. A similar explanation has been hypothesized for other drug-induced myoclonus, such as lithium and fluoroquinolones [26,27]. Notably, these drugs share case presentation similarities regarding myoclonus and progressive cognitive impairment, known by some authors as Creutzfeldt–Jakob-like syndrome.

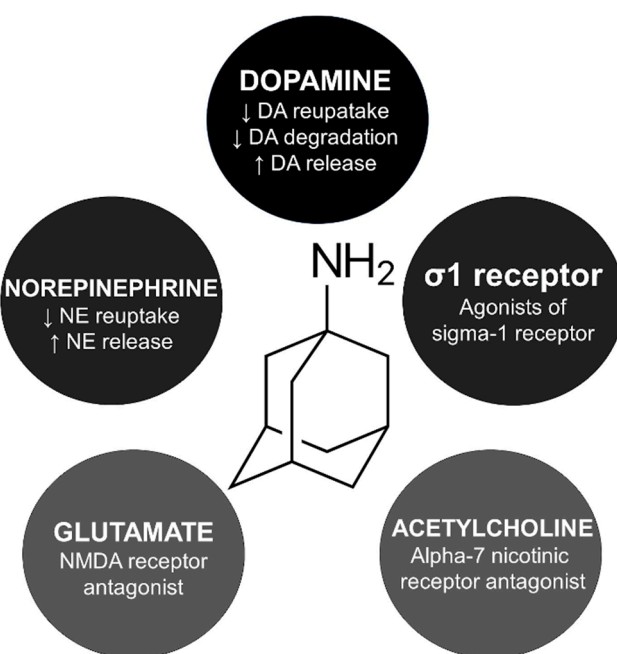

**Figure 1.** Mechanism of action and skeletal formula of amantadine. DA, dopamine; NE, norepinephrine; NMDA, N-methyl-D-aspartate.

We searched six databases to locate the studies on amantadine and myoclonus published from 1980 to June 2022 in electronic form. Excerpta Medica (Embase), Google Scholar, Latin American & Caribbean Health Sciences Literature (Lilacs), Medline, Scientific Electronic Library Online (Scielo), and Science Direct were searched. Search terms were "myoclonus, movement disorder". These terms were combined with "amantadine, 1-adamantylamine" (Table S1). Publications in English were included in the search (Table 1) [2,4–18].

The most common MCL presentation was multifocal jerks in the limbs, but focal jerks were also observed. In this context, some cases that described cranial myoclonus, also known as branchial or vocal myoclonus, involving facial muscles were misdiagnosed as stuttering [4,10]. Notably, amantadine is a type of NMDA antagonist, like ketamine and phencyclidine, which are well known to cause a head-twitch response in rat models [28]. Therefore, the association of myoclonus with amantadine therapy is expected. Notably, amantadine was prescribed for many different types of disorder (e.g., Parkinson's disease, progressive supranuclear palsy, disorders of consciousness, and depression), so differences from a pathophysiological point of view on the role of amantadine in these very different conditions is possible.

Amantadine-induced myoclonus was slightly more common in men (12/23). The population affected was the elderly, with a mean and median age of 67.7 (SD: 9.8) and 64 years (age range: 53–87). It is worth mentioning that the studied population involved patients with Parkinson's disease, which can explain the age of the people affected [29].

Most of the cases were not reported by movement disorder specialists, which could have led to possible misdiagnosis of the abnormal movement in some reports. In only 43% (10/23) of the cases, electrodiagnostic studies were performed. The description of supporting studies is essential for defining myoclonus sources [30]. The most frequent source of myoclonus was cortical, but some authors reported a subcortical origin.

**Table 1.** Clinical reports of myoclonus associated with amantadine.

| Reference (Year) | Age/Sex | AMT Dosing (mg/Daily) and Indication | MCL Presentation | KF | MCL Onset [a] | Management | MCL Recovery [b] | EEG [c] | F/U | Considerations |
|---|---|---|---|---|---|---|---|---|---|---|
| Chevalier et al. (1980) [2] | 64, M | NA; PD | Generalized MCL. | N | NA | AMT withdrawal | NA | NA | NA | First report of AMT-induced MCL. Diuretics increased the intoxication by AMT. |
| Pfeiffer et al. (1996) [4] | 64, F | 200; PD | Focal (cranial) MCL. | N | NA | AMT withdrawal. Clonazepam was attempted. | NA | NA | NA | First report of vocal (cranial) MCL. Misdiagnosed with stuttering. Videotape. |
| Matsunaga et al. (2001) [5] | 87, F | 100; NA | Generalized (multifocal) MCL. Cortical MCL. | Y | 30 days | AMT withdrawal | 14 days | Abnormal | CR | Plasma AMT concentration. |
| | 78, F | 200; PD | Generalized (multifocal) MCL. Cortical MCL. | N | 90 days | AMT withdrawal | 8 days | Abnormal | CR | Dose-dependent MCL. AMT-dose increase was associated with a rise in MCL frequency. |
| | 79, F | 200; PD | Generalized (multifocal) MCL. Cortical MCL. | Y | 9 days after worsening of renal function | AMT withdrawal | 7 days | Abnormal | | Plasma AMT concentration. MCL appeared with worsening renal function. |
| Nakata et al. (2006) [6] | 70, F | 150; PD | Generalized MCL. | Y | NA (y) | AMT withdrawal | 21 days | Normal | CR | Plasma AMT concentration. |
| | 74, F | 200; Depression | Generalized MCL | N | NA (y) | AMT withdrawal | 21 days | Normal | CR | Possible serotonin syndrome. |
| | 73, F | 300; PD | Generalized MCL. | Y | 7 days | AMT withdrawal | NA | Abnormal | No | Possible serotonin syndrome. |
| Cheng et al. (2008) [7] | 78, M | 100; PD | Generalized MCL | Y | 3 days | AMT withdrawal | 12 days | Abnormal | CR | Serotonin syndrome. |
| Hong et al. (2008) [8] | 59, F | 200; PD | Generalized MCL | Y | 11 days | AMT withdrawal | NA | NA | NA | Possible interaction with pramipexole. |
| Nishikawa et al. (2009) [9] | 62, F | 200; PD | Generalized MCL | Y | NA | AMT withdrawal | NA | NA | CR | Plasma AMT concentration. |
| | 55, F | 150; PD | Generalized MCL | Y | NA | AMT withdrawal | NA | NA | CR | Plasma AMT concentration. |
| Gupta et al. (2010) [10] | 63, M | 300; parkinsonism with postural instability | Focal (cranial) MCL. Resting and action MCL of lower face. | N | NA (several months) | AMT withdrawal | NA | NA | CR | Videotape. Misdiagnosed as stuttering. |
| Hardwick et al. (2010) [11] | 63, M | NA; pruritus | Generalized MCL | Y | NA | AMT withdrawal | 56 days | Normal | CR | Plasma AMT concentration. |
| Yarnall et al. (2012) [12] | 74, M | 200; PSP | Generalized MCL | N | 26 days | AMT withdrawal | 5 days | NA | CR | PSP diagnosis supported by abnormal DaTSCAN. |

**Table 1.** *Cont*.

| Reference (Year) | Age/Sex | AMT Dosing (mg/Daily) and Indication | MCL Presentation | KF | MCL Onset [a] | Management | MCL Recovery [b] | EEG [c] | F/U | Considerations |
|---|---|---|---|---|---|---|---|---|---|---|
| Kawamura et al. (2013) [13] | 58, M | NA; PSP | Generalized MCL | N | NA | Clonazepam was attempted | NA | NA | NA | Giant potential was found in somatosensory evoked potential of the median nerve. |
| Estraneo et al. (2015) [14] | 57, F | 200; coma state | Focal (cranial) MCL | N | 21 days | AMT withdrawal | 21 days | Abnormal | NA | Three attempts of AMT rechallenge. |
| Janssen et al. (2017) [15] | 66, M | 300; PD with Levodopa-induced dyskinesias | Generalized MCL | N | 30 days | AMT withdrawal | 14 days | NA | CR | Videotape. |
| Kunieda et al. (2017) [16] | 83, M | 150; PD | Generalized MCL. | Y | 5 days | AMT withdrawal | 29 days | NA | CR | Plasma AMT concentration |
| | 53, M | 100; spontaneity | Generalized MCL. | Y | 21 days | AMT withdrawal. AMT rechallenge. | NA | NA | CR | AMT rechallenge without symptoms occurrence. |
| Dames et al. (2020) [17] | 55, M | 400; PD | Generalized MCL. | Y | NA (y) | AMT withdrawal | 7 days | NA | No | Videotape. |
| Poon et al. (2021) [18] | 80, M | PD with Levodopa-induced dyskinesias | Generalized MCL. Asterixis. | N | 9 days | AMT withdrawal | 3 days | NA | CR | Subcortical MCL. |
| Present report | 64, M | PD | Generalized MCL. Asterixis. | N | 7 days | AMT withdrawal | 3 days | Normal | CR | Subcortical MCL. |

Abbreviations: AMT, amantadine; CR, complete recovery; EEG, electroencephalography; F, female; F/U, follow-up; KF, kidney failure/renal dysfunction; M, male; MCL, myoclonus; N, no; NA, not available/not reported; PD, Parkinson's disease; PSP, progressive supranuclear palsy; Y, yes; y, years. [a] MCL onset: time from AMT starting until the MCL onset. [b] MCL recovery: time from AMT withdrawal (management) until MCL recovery. [c] Some of the EEG abnormalities were increased predominant background alpha activity and intermittent generalized diffuse slow waves.

In pharmacokinetic studies, the plasma concentration of amantadine ranged from 100 to 2000 ng/mL [31]. In elderly individuals, 1000 to 2000 ng/mL is considered dangerous by some authors due to a higher incidence of side effects such as hallucinations and delirium [31,32]. We reviewed the literature and provide a figure about the amantadine concentration (Figure 2) [5,6,9,11,16,33,34]. Interestingly, all the cases that reported myoclonus had amantadine concentrations above 3000 ng/mL [9]. Despite prevailing renal elimination, the metabolism of amantadine is not yet fully clarified because 5–15% of an oral dose is apparently acetylated and acetylator phenotype might influence toxicity [35].

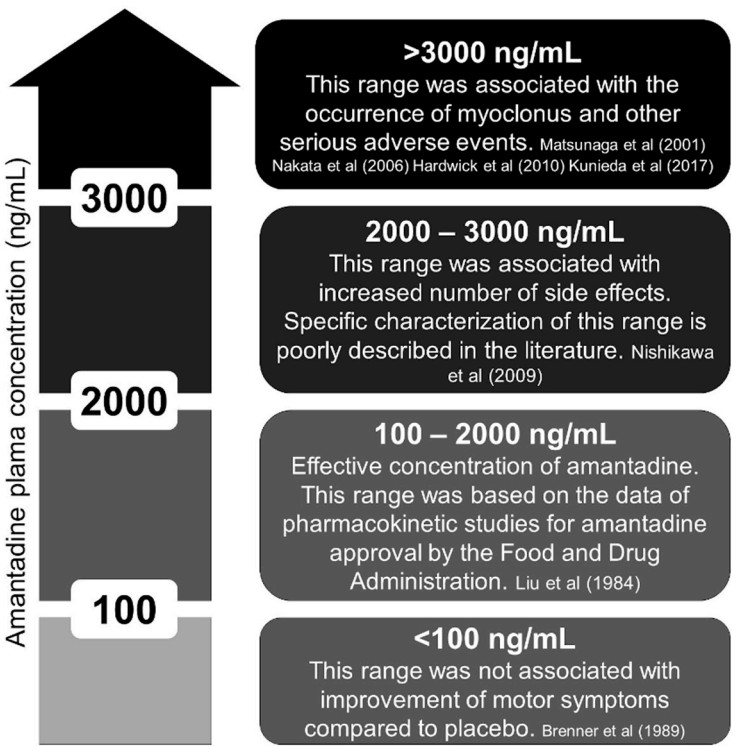

**Figure 2.** Plasma concentration of amantadine (ng/mL). Ineffective, effective, unknown, and toxic ranges [5,6,9,11,16,33,34].

The availability and costs regarding the measurement of serum amantadine levels are still a limitation to a specific approach to the adverse events associated with this medication. In this context, developing extended-release formulations with late peak plasma concentration and a longer half-life may be associated with increased side effects. Thus, assessing drug levels for adequately managing parkinsonism will be mandatory. Meanwhile, clinicians should rely on the Cockcroft–Gault formula to estimate creatinine clearance as a risk factor for developing side effects related to amantadine. Noteworthy, levodopa, considered the mainstay of treating Parkinson's disease, was already associated with myoclonus. However, levodopa-induced myoclonus is a relatively late complaint because most individuals only present this side effect after ten years of levodopa use [36].

Eighty to ninety percent of amantadine is excreted unchanged by glomerular filtration and tubular secretion. In this way, renal dysfunction can cause accumulation of this drug in several organs, such as the lungs and kidneys. Interestingly, the approximate half-life of amantadine is sixteen hours in individuals with normal renal function and eight days in dialytic individuals [31]. Therefore, we analyzed the data of the cases regarding renal impairment in the individuals reported in Table 1. Of the 23 subjects, 12 had at least mildly decreased renal function. However, some authors did not describe creatinine levels or creatinine clearance.

Dames et al. reported a case of a patient falling several times a day for years. The authors reported a possible association between falling and amantadine therapy. In this

context, they observed that the patient had generalized myoclonus contributing to his imbalance. The amantadine clinical trials revealed an increased dose-dependent percentage of falls in patients with Parkinson's disease [37,38]. GOCOVRI's trials for levodopa-induced dyskinesia demonstrated a higher incidence of falls with amantadine. This finding was mainly observed in patients over 65 years old [39].

## 4. Conclusions

Myoclonus secondary to amantadine has rarely been reported in the literature. The pathophysiology of this association is probably related to serotoninergic pathways. Clinicians should consider amantadine-induced myoclonus as a cause of falling in Parkinson's disease patients. Future reports should describe electrodiagnostic studies for a determination of the myoclonus source.

**Supplementary Materials:** The following supporting information can be downloaded at: https://www.mdpi.com/article/10.3390/clinpract13040075/s1, Video S1: Asterixis, Table S1: FreeText and MeSH search terms in the US National Library of Medicine.

**Author Contributions:** J.P.R. and A.L.F.C. conceived and designed the methodology of the literature review. J.P.R. and A.L.F.C. extracted and collected the relevant information and drafted the manuscript. A.L.F.C. supervised the article selection and reviewed and edited the manuscript. J.P.R. and A.L.F.C. reviewed and edited the manuscript. All authors have read and agreed to the published version of the manuscript.

**Funding:** This research received no external funding.

**Institutional Review Board Statement:** Not applicable.

**Informed Consent Statement:** Written informed consent has been obtained from the patient to publish this paper.

**Data Availability Statement:** Not applicable.

**Conflicts of Interest:** The authors declare no conflict of interest.

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
