# Peer review of "Myoclonus Secondary to Amantadine: Case Report and Literature Review"

_clinpract, doi:10.3390/clinpract13040075_

Round 1

Reviewer 1 Report

An interesting case with literature review. I have some minor suggestions:

1) Livedo reticularis is a common side effect of amantadine. Please add. 

2) Table 1 is unnecesary. The results of laboratory data could be commented in the text. 

3) It should be commented data on levodopa-induced myoclonus in the discussion. 

4) The authors should consider to mention a review on Drug-induced myoclonus (PMID 14728056.

Author Response

An interesting case with literature review. I have some minor suggestions: 1) Livedo reticularis is a common side effect of amantadine. Please add.

Authors: The authors included this information in the following structure, “The most common adverse events of amantadine are dizziness, dry mouth, peripheral edema, and livedo reticularis.”

2) Table 1 is unnecesary. The results of laboratory data could be commented in the text.

Authors: Table 1 was removed as requested by the Reviewer.

3) It should be commented data on levodopa-induced myoclonus in the discussion. 4) The authors should consider to mention a review on Drug-induced myoclonus (PMID 14728056.

Authors: The authors would like to thank the reviewer for the idea that improved the quality of the manuscript. The following structure was added, “Noteworthy, levodopa, considered the mainstay of treating Parkinson's disease, was already associated with myoclonus. But, levodopa-induced myoclonus is a relatively late complaint because most individuals only present this side effect after ten years of levodopa use.”

Reviewer 2 Report

The paper by Pitton Rissardo and Fornari Caprara is an interesting case report associated with a literature review on the possible link between amantadine and the onset of myoclonus. The Authors conducted an extensive literature search and well described pharmacological and molecular mechanisms in which this drug is involved. The paper is well written, but I have a few observations for the Authors: 

- The patient described in their case report has (presumably, since cut-offs are not reported) increased protein values in the CSF. How was this justified by the Authors? Could there be a link with the onset of myoclonus? 

- Why did the Authors not think of carrying out an in-depth neurophysiological study, e.g. search for potential giants in SEPs, back-averaging,...? 

- With regard to the cases described in the literature, many different types of indications are lumped together (e.g. PD, PSP, disorders of consciousness, depression, etc.). I suggest that the Authors highlight, if possible, differences from a pathophysiological point of view on the role of amantadine in these very different conditions. Furthermore, the Authors could describe more precisely what is meant by abnormal EEG and potential links of these abnormalities with other available information

Author Response

The paper by Pitton Rissardo and Fornari Caprara is an interesting case report associated with a literature review on the possible link between amantadine and the onset of myoclonus. The Authors conducted an extensive literature search and well described pharmacological and molecular mechanisms in which this drug is involved. The paper is well written, but I have a few observations for the Authors: - The patient described in their case report has (presumably, since cut-offs are not reported) increased protein values in the CSF. How was this justified by the Authors? Could there be a link with the onset of myoclonus?

Authors: The CSF findings were “Cerebrospinal fluid analysis showed 60 mg/dL of glucose (98 mg/dL plasma glucose), 30 mg/dL protein, 0 leukocytes, and 0 red blood cells.” In the hospital institution where the laboratory exam was performed, provide the following normal range for CSF protein 15-45 mg/dL. Therefore, the authors believe that the CSF protein results were not abnormal.

- Why did the Authors not think of carrying out an in-depth neurophysiological study, e.g. search for potential giants in SEPs, back-averaging,...?

Authors: The authors greatly appreciate the idea of the Reviewer. Providing more specific data regarding this side effect would be extremely significant, especially neurophysiological information. But, there are important limitations in the public health system of the author’s country, like equipment availability and funding to support further evaluations.

- With regard to the cases described in the literature, many different types of indications are lumped together (e.g. PD, PSP, disorders of consciousness, depression, etc.). I suggest that the Authors highlight, if possible, differences from a pathophysiological point of view on the role of amantadine in these very different conditions. Furthermore, the Authors could describe more precisely what is meant by abnormal EEG and potential links of these abnormalities with other available information.

Authors: The authors included the following structure to address this comment, “Noteworthy, amantadine was prescribed for many different types of disorders (e.g., Parkinson's disease, progressive supranuclear palsy, disorders of consciousness, and depression), so differences from a pathophysiological point of view on the role of amantadine in these very different conditions is possible.” Also, we re-reviewed the literature to include the following information regarding EEG abnormalities, “Some of the EEG abnormalities were increased predominant background alpha activity and intermittent generalized diffuse slow waves.”

Reviewer 3 Report

At the end of the abstract, it is necessary to add information about the discussion and the conclusions.

In the discussion, the figure is added. However, its purpose is not clear. It is briefly mentioned that the mechanism of amantadine to control Parkinson's is unknown, and about five indirect pathways were already studied to explain amantadine's involvement in the cortico-striato-pallido-thalamo-cortical loop. This part should be deleted or described in more detail.

In conclusion, the pathophysiology of this association is probably related to serotonergic pathways. However, the discussion did not show sufficient evidence to conclude this. It's not strictly a conclusion; it's part of the discussion.

NA

Author Response

At the end of the abstract, it is necessary to add information about the discussion and the conclusions.

Authors: The authors believe that the present format of the abstract describes as much information as possible. There is a limit of words in the abstract to 200 words, which was done. Also, the abstract describes information regarding the reported case and a summary of the results from the literature.

In the discussion, the figure is added. However, its purpose is not clear. It is briefly mentioned that the mechanism of amantadine to control Parkinson's is unknown, and about five indirect pathways were already studied to explain amantadine's involvement in the cortico-striato-pallido-thalamo-cortical loop. This part should be deleted or described in more detail.

Authors: The authors deleted the figure as requested by the Reviewer.

In conclusion, the pathophysiology of this association is probably related to serotonergic pathways. However, the discussion did not show sufficient evidence to conclude this. It's not strictly a conclusion; it's part of the discussion.

Authors: The authors deleted this information in the discussion to better understand this chapter.

Round 2

Reviewer 2 Report

The Authors appropriately responded to comments and suggestions. The paper is now, in my opinion, suitable for publication